# Isolation, identification and antibacterial activity of endophytes from the seeds of *Panax japonicus*

Rui Jin[1,2*], Tingting Tang[1,2*], Xilun Huang[1,2], Juan Huang[2], E. Liang[2], Lai Zhang[ID][2*]

**1** School of Life Science, Guizhou Normal University, Guizhou, Guiyang, China, **2** Anshun University/ Innovation Center for Efficient Agricultural of Guizhou Mountain Characteristics, Guizhou, Anshun, China

* 975575681@qq.com (LZ); 1657943467@qq.com (RJ); 2021172860@qq.com (TT)

## Abstract

The purpose of this study is to isolate and identify endophytes from the seeds of *Panax japonicus*, find growth-promoting endophyte that favors seed germination in *P. japonicus*, and provide theoretical reference for further research on the effect of endophytes on seed germination.

## Methods

The streak plate method isolated the endophytic bacteria from the seeds. The isolated endophytes were identified by morphology, ITS and 16S rDNA sequence analysis. The antibacterial activity of the isolated endophytes and their metabolites was studied by disk method.

## Results

The seed was disinfected with 75% alcohol for 30s and 1% $HgCl_2$ for 9 min, and the contamination rate was the lowest (3.33%). Seven endophytic fungi strains were isolated from the seeds of *P. japonicus*, named *Pj*Z1, *Pj*Z2, *Pj*Z3, *Pj*Z4, *Pj*Z5, *Pj*Z6, and *Pj*Z7, respectively, four strains belonged to the genus *Fusarium tricinctum,* one to *Fusarium reticulatum,* one to *Fusarium sarcochroum,* and one to *Alternaria alternata*; Three strains of endophytic bacteria were isolated and named *Pj*X1, *Pj*X2 and *Pj*X3, respectively. After morphological observation and 16S rDNA gene sequencing, two strains belong to the genus *Enterobacteriaceae bacterium* and one belongs to the genus *Pseudomonas sp*. Among the endophytic fungi strains isolated, only *Pj*Z4 showed significant bacteriostatic activity against *Escherichia coli*, *Pj*Z3, *Pj*Z4, *Pj*Z5 and *Pj*Z7 had antibacterial activity against *Bacillus subtilis*, and none of the 7 strains had antibacterial activity against *Staphylococcus aureus*. It was found that the metabolites of 7 endophytic fungi and 3 endophytic bacteria had no antibacterial activity.

**Data availability statement:** All relevant data are within the paper and its Supporting Information files.

**Funding:** Science and Technology Funded Project of Guizhou Provincial Science and Technology Department (Guizhou Science and Technology Foundation -ZK Grant NO.(2023) Key 002). National Natural Science Foundation of China (Grant NO.31660252). Guizhou Provincial Department of Education Youth Science and Technology Talent Development Project (Qian Jiao He KY character Grant NO.[2022]029). Anshun College graduate Research Innovation project asxyyjscx Grant NO.202410. The funders had no role in study design, data collection and analysis, decision to publish, or preparation of the manuscript.

**Competing interests:** I have read the journal's policy and the authors of this manuscript have the following competing interests:No conflict of interest exits in the submission of this manuscript, and manuscript is approved by all authors for publication.I would like to declare on behalf of my co-authors that the work described was original research that has not been published previously, and not under consideration for publication elsewhere, in whole or in part. All the authors listed have approved the manuscript that is enclosed.

## Conclusion

In this study, we investigated the method of *P. japonicus* seed disinfection, and the results showed that 75% alcohol disinfection for 30s and 1% $HgCl_2$ disinfection for 9 min resulted in the lowest contamination rate of 3.33%, in which the greatest influence on the seed disinfection effect was the concentration of $HgCl_2$. Meanwhile, 10 endophyte species of *P. japonicus* seeds were isolated and identified, *Fusarium spp.* and *Alternaria alternata*. were not the dominant species in the growth and development process of *P. japonicus* while *Enterobacteriaceae* and *Pseudomonas sp.* were the growth-promoting endophyte in the promotion of plant growth and development, and provide a theoretical reference for further research on the biological functions and active substances of endophyte in the seeds of *P. japonicus*.

## Introduction

*Panax japonicus* C.A. Meyer. is a famous folk medicine of the Tujia ethnic group, and is regarded as the "king of herbs' by the Tujia ethnic group of Enshi Hubei. The chemical composition mainly includes saponins, sugars, amino acids, volatile oils, flavonoids, nucleosides and inorganic salts, etc. [1]. Pharmacological studies have shown that *P. japonicus* has a good therapeutic effect in enhancing immunity, anti-fatigue, cardioprotection, anti-tumor, anti-myocardial ischemia, sedation and analgesia, and treatment of rheumatoid arthritis [2–4]. Because of the high efficacy and high price of the rare wild Chinese herbal medicine, it is almost difficult to find wild resources of *P. japonicus* in the wild, so the artificial propagation of *Panax japonicus* seeds is urgent [5]. Therefore, the selection of *P. japonicus* seeds, leaves and stems for seedling propagation is a better choice of explants. Notably, the seeds of *P. japonicus* have the phenomenon of after-ripening, and the contamination rate is high during the histoculture's germination process, making germination difficult. Therefore, we carried out further exploration. It has been shown that endophytes of medicinal plants are important for promoting the growth of medicinal plants, improving the quality of medicinal plants, and eliminating the barriers of continuous cultivation [6]. Scholars have isolated endophytic bacteria from roots, stems, leaves, fruits, and rhizomes of *P. japonicus* [7]. However, the isolation of endophytic fungi and bacteria from the seeds of *Panax japonicus* has not yet been reported. Therefore, the isolation of endophytes from the seeds of *P. japonicus* and their classification, characterization, and bacteriostatic studies can provide certain theoretical references for the further development of the research on the biological functions and active substances of the endophyte in the seeds of *P. japonicus*, as well as for the germination of the seeds of *P. japonicus*.

## Materials and methods

### Plant material

Fresh seeds of *P. japonicus* were obtained from the experimental site of *P. japonicus* cultivation in the "Guizhou Innovation Center of Mountain Specialty and Efficient Agriculture" of Anshun College, Anshun City, Guizhou Province, China.

## Culture medium

Fungi isolation medium: Potato Dextrose Agar Medium (PDA solid medium) 40.1g, distilled water 1000mL, pH:5.8. Bacterial isolation medium: Beef Extract Peptone Medium (NA solid medium): Peptone 10g, Beef Extract 3g, Sodium Chloride 5g, Agar 20g, distilled water 1000mL, pH:7.2.

Experimental Medium for Fungi Inhibition Nutrient broth (NB liquid medium): peptone 10g, Beef Extract 3g, sodium chloride 5g, distilled water 1000mL, pH: 7.2, Beef Extract Peptone Medium (NA solid medium). Fungi fermentation medium: PDB liquid medium, Beef Extract Peptone Medium (NA solid medium), Nutrient Broth (NB liquid medium). PDA, NA and NB medium are prepared as above.

Bacterial fermentation medium: LB solid medium: LB dry powder 25g, distilled water 1000mL, Agar 15g pH:7.0; LB liquid medium: LB dry powder 25g, distilled water 1000mL, pH:7.0; MH medium: Mueller-Hinton Agar 38g, distilled water 1000mL, pH: 7.3.

## Bacterial strain

Strains *Staphylococcus aureus*, *Escherichia coli*, *Bacillus subtilis*, purchased from Guangdong Huankai Biotechnology Co.

## Screening of seed disinfection conditions

With 75% alcohol disinfection time of 30s, 60s, and 90s, $HgCl_2$ concentration of 0.1%, 0.5%, and 1.0%, $HgCl_2$ disinfection time of 3min, 6min, and 9min as the influencing factors, a 3-factor, 3-level orthogonal experiment was designed according to the orthogonal table of $L_9$ $(3^3)$, with a total of 9 treatments [8]. The results of the experiment were analyzed using the software SPSS.

## Disinfection

Before the isolation of endophytes from the seeds of *P. japonicus*, the seeds of *P. japonicus* were disinfected under different conditions and the contamination rate under different disinfection conditions was analyzed. The seeds were rinsed under running water and then disinfected with the above orthogonal design combinations. After disinfection, the seeds were inoculated on an MS medium: 4.47g MS dry powder,1000mL $H_2O$, Agar 5.5g pH:5.8, and the contamination rate was statistical.

## Isolation and purification of endophytic fungi

Take the fresh seeds of *P. japonicus* and wash them under running water, then rinse them 1–3 times with sterile water, disinfect them with 75% alcohol for 30s, 1% $HgCl_2$ for 9min, rinse them with sterile water for 2–3 times and then dry them with sterile filter paper of 5cm² in size, peel them and then half-cut, put them on PDA medium with the cut side facing down [9], and then place them in an incubator at 28°C for cultivation. After the growth of mycelium, pick the mycelium for repeated isolation and purification 2–3 times for preservation of the isolated strains.

## Isolation and purification of endophytic bacteria

Take fresh seeds of *P. japonicus* and rinse them under running water, then rinse them 1–3 times with sterile water, same as above for seed disinfection, peel the seeds and cut them in half, place them on NA medium with the cut side facing down, and place them in the incubator at 28°C for cultivation. After the growth of colonies, the strains were purified using the streak plate method, the strains were purified 2–3 times and then preserved for isolated line purification.

## Identification of endophytes

**Morphological observation.** Picked the purified fungi hyphae and made them into makeshift slices, observed and recorded (hyphae color, hyphae morphology, spore morphology, etc.) through an optical microscope, and referred to

"Fungi Identification Manual" [10] for preliminary identification of fungi. (Three replications were performed for each strain). The purified bacterial strains were subjected to continuous observation, and records were made (diameter size, color, morphology, dryness, whether transparent or not, etc. of the colonies). Picked single colonies to make makeshift slices for Gram staining, and refer to the "Manual of Identification of Common Bacterial Systems" for preliminary analysis and identification [11].

**Molecular biology identification:** Endophytic fungi DNA was extracted according to the instructions of the fungal DNA extraction kit (Solarbio), and endophytic bacterial DNA was extracted according to the instructions of the bacterial DNA extraction kit. Subsequently, extracted DNA was used as a template for PCR amplification respectively.

Endophytic fungi were amplified by PCR using the universal primers of fungi, ITS1 (5'-TCCGTAGGTGAACCTG CGG-3') and ITS4 (5'-TCCTCCGCTTATTGATATGC-3') as primers, and the total volume of the PCR reaction system was 50 μL, containing:4 μL of DNA template, 2 μL of each of ITS1 and ITS4, and 2×MasterMix 25 μL, and add ddH$_2$O 17 μL. PCR reaction conditions: pre-denaturation at 95°C for 5 min, 95°C for 30s, annealing to 58°C and holding for 30s, followed by extension at 72°C for 80s for 35 cycles, and finally extension at 72°C for 10 min and storage at 4°C.

Bacterial were amplified by PCR using the universal primers 27F (5'-AGAGTTTGATCCTGGCTCAG-3') and 1492R (5'-TACGGCTACCTTGTTACGACTT-3') of bacteria as primers, and the PCR reaction system was 50 μL: 2 μL of DNA template, 1 μL each of 27F and 1492R, 2×MasterMix 25 μL, and add ddH$_2$O to 50 μL. PCR reaction conditions: pre-denaturation at 95°C for 5 min, 95°C for 30s, annealing to 55°C for 30s, followed by extension at 72°C for 1 min for 35cycles, and finally extension at 72°C for 10 min, and storage at 4°C.

The primer concentrations mentioned above were all 10 μmol.

After the PCR amplification reaction, the products were detected by the ratio is 1% agarose gel electrophoresis. The amplification products were sequenced by Sangon Biotech. The sequencing results were logged into the official website of NCBI for BLAST comparison analysis, and MEGA11 software was used to create a phylogenetic tree to identify the endophytes.

## Determination of bacteriostatic activity of endophytic fungi and endophytic bacterial metabolites

**Determination of bacteriostatic activity of endophytic fungi.** The pure indicator strains (*Staphylococcus aureus, Bacillus subtilis, Escherichia coli*) were picked and inoculated onto LB medium by streak plate method for overnight cultivation; picked single colonies on LB medium again for plate streaking overnight culture; pick single colonies to 0.85% saline to prepare the bacterial solution, adjust the concentration of the bacterial solution and measured by UV-visible spectrophotometer (UV-5500, Shanghai, China), until OD$_{600}$=0.2, and the bacterial solution was dipped into cotton swabs to evenly spread onto NA medium to make the bacterial plate containing the indicator strains; a perforator with a diameter of 0.7 cm was used to prepare fungi chunks were prepared on the endophytic fungi colonies with a 0.7 cm diameter punch, and placed on the fungi plate containing the indicator species, and 3 chunks were inoculated equidistant in each plate; 3 groups of replicates were made for each endophytic fungi, and the no antibiotics blank drug-sensitive paper(Shandong Best Microbial Technology Co., Ltd.) was used as the blank control, and the culture was incubated at a constant temperature of 28 °C, and the bacterial inhibitory activity was observed after 3 d. The diameter results were summarized as follows, the size of the ring of inhibition was recorded and the average value was calculated.

**Determination of bacteriostatic activity of metabolites of endophytic fungi.** The endophytic fungi strain was transferred to a 100 mL conical flask containing 50 mL of PDB liquid medium; the fermentation culture was oscillated at 28 °C and 150 r/min for 7 d. The supernatant was centrifuged at 4 °C and 12000 r/min for 10 min and then filtered through a 0.22 μm sterile filter membrane, and the fermentation broth was obtained as the metabolites of the endophytic fungi. The blank drug-sensitive paper was placed in the bacterial plate, and each plate was inoculated with 3 pieces of filter paper at equal distance, and 15 μL of fermentation broth (endophytic fungal metabolites) was added to each piece of filter paper dropwise by pipettor; 3 groups of replicates were set up for each endophytic fungi metabolites, and the blank

drug-sensitive paper was used as the blank control, and the culture was incubated at a constant temperature of 28 °C, and whether there was any circle of inhibition was observed and the diameter size of the circle of inhibition was measured after 3 d, and the mean value was taken for calculation.

**Determination of bacteriostatic activity of endophytic bacterial metabolites.** Pick a small amount of pure endophytes, inoculate them into fresh LB medium by plate streaking method, activate and cultivate them at 30°C for 12~16h. Pick the activated single colony and transfer it to LB liquid medium, incubate it at 30°Cand 230r/min overnight; take 2 mL of the seed solution into 50 mL of the newly formulated LB liquid medium, incubate it at 30°Cand 180 r/min for 7 d, and then centrifuge the supernatant after centrifugation for 5 min at 8000 r/min, and filter the fermentation broth obtained by filtration with 0.22μm sterile filter membrane to remove bacteria. After centrifugation, the supernatant was filtered with a 0.22μm sterile membrane to remove bacteria, and the fermentation broth obtained from the filtration was endophytic bacterial metabolites. Determination of bacteriostatic activity of endophytic bacterial metabolites was the same as above for the determination of bacteriostatic properties of endophytic fungi metabolites.

## Results and analysis

### Results of an orthogonal test on seed disinfection of *P. japonicus*

The results of the effects of different disinfection time and disinfectant concentrations on the disinfection effect of *P. japonicus* seeds showed that 75% alcohol disinfection for 30 s and 1% $HgCl_2$ disinfection for 9 min resulted in the lowest contamination rate of 3.33% (Table 1), and the analysis of extreme variance (ANOVA) indicated that the three factors had the greatest effect on the disinfection effect of *Panax japonicus* seeds as $HgCl_2$, followed by disinfection time and finally 75% alcohol; ANOVA pointed out that the differences between the set ANOVA indicated that the differences among 75% alcohol, $HgCl_2$ and disinfection time on the contamination rate of *P. japonicus* seeds were not significant at the statistical level within the set gradient range. However, it was obtained that $HgCl_2$ (%)> time (min) > 75% alcohol affected seed disinfection in the order of priority (Tables 2 and 3).

### Isolation and purification of endophyte from the seeds of *P. japonicus*

Seven endophytic fungi colonies, named *PjZ*1, *PjZ*2, *PjZ*3, *PjZ*4, *PjZ*5, *PjZ*6, *PjZ*7(Their NCBI accession numbers are PX064304, PX064305, PX064306, PX064307, PX064308, PX064309, and PX064310), and three endophytic bacterial colonies, named *PjX*1, *PjX*2, and *PjX*3(Their NCBI accession numbers are PX064093, PX064094, PX064095), were isolated from the seeds of *P. japonicus.*The external morphology and micromorphology of fungi and bacterial are shown in Figs 1 and 2.The morphology of all three endophytic bacteria was round, with neat edges and slightly raised centers,

**Table 1. Results of the orthogonal test of seed disinfection of *P. japonicus*.**

| Deal with | Factor | | | Results statistics | | |
|---|---|---|---|---|---|---|
| | 75% alcohol (s) | $HgCl_2$ (%) | Time (min) | Number of inoculations | Contamination | Contamination rate (%) |
| 1 | 30 | 0.1 | 3 | 30 | 10 | 33.33 |
| 2 | 30 | 0.5 | 6 | 30 | 2 | 6.67 |
| 3 | 30 | 1.0 | 9 | 30 | 1 | 3.33 |
| 4 | 60 | 0.1 | 6 | 30 | 18 | 60.00 |
| 5 | 60 | 0.5 | 9 | 30 | 4 | 13.33 |
| 6 | 60 | 1.0 | 3 | 30 | 6 | 20.00 |
| 7 | 90 | 0.1 | 9 | 30 | 7 | 23.33 |
| 8 | 90 | 0.5 | 3 | 30 | 14 | 46.67 |
| 9 | 90 | 1.0 | 6 | 30 | 4 | 13.33 |

**Table 2. Analysis of variance (ANOVA) of the factors affecting seed disinfection of *P. japonicus*.**

|  | Type III sum of squares | Degree of freedom | Mean square | F | Significance |
|---|---|---|---|---|---|
| Alcohol | 466.667 | 2 | 233.333 | 0.677 | 0.596 |
| HgCl$_2$ | 1088.844 | 2 | 544.422 | 1.58 | 0.388 |
| Time | 622.444 | 2 | 311.222 | 0.903 | 0.525 |
| Inaccuracies | 688.978 | 2 | 344.489 |  |  |

**Table 3. Extreme variance analysis of factors affecting seed disinfection of *P. japonicus*.**

| Item | 75% alcohol (s) | HgCl$_2$ (%) | Time (min) |
|---|---|---|---|
| K1 | 43.33 | 116.67 | 100.00 |
| K2 | 93.33 | 66.67 | 80.00 |
| K3 | 83.33 | 36.67 | 40.00 |
| k1 | 14.44 | 38.89 | 33.33 |
| k2 | 31.11 | 22.22 | 26.67 |
| k3 | 27.78 | 12.22 | 13.33 |
| R | 16.67 | 26.67 | 20.00 |
| Order of priority | HgCl$_2$ (%)> time (min) > 75% alcohol | | |

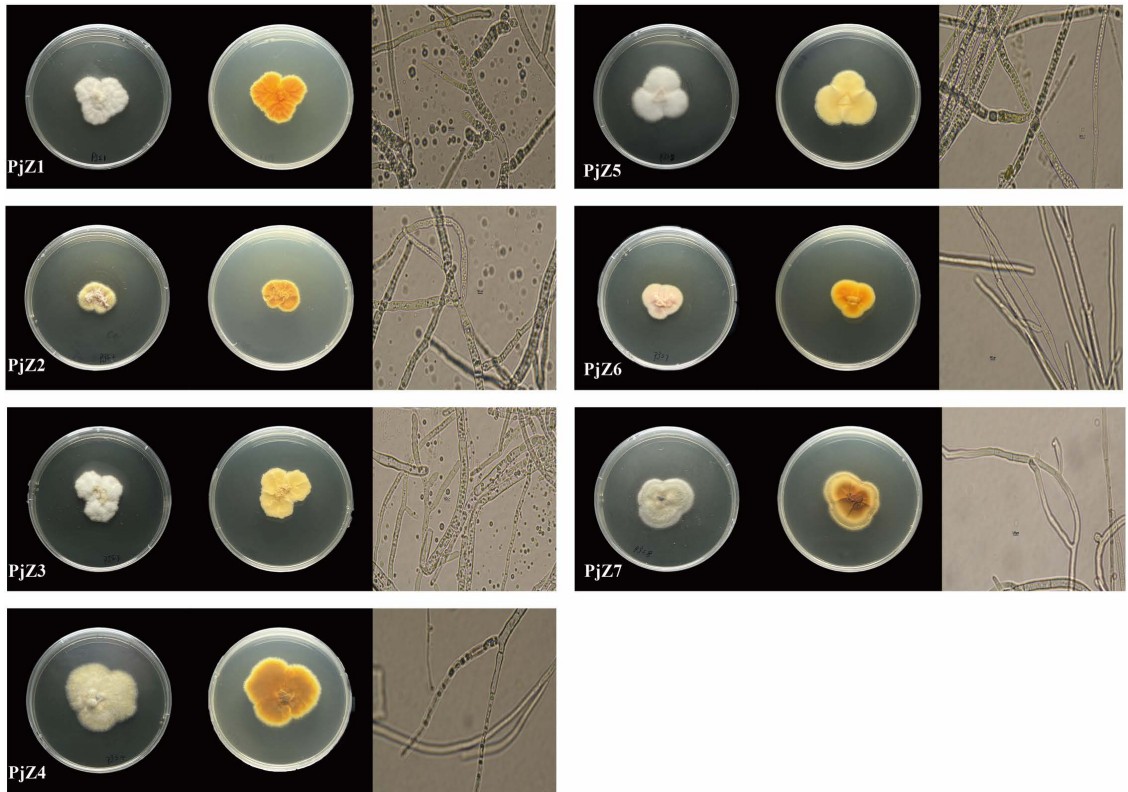

**Fig 1. Characterization of colonies of the endophytic fungi *PjZ1-PjZ7* in the seeds of *P. japonicus*.**

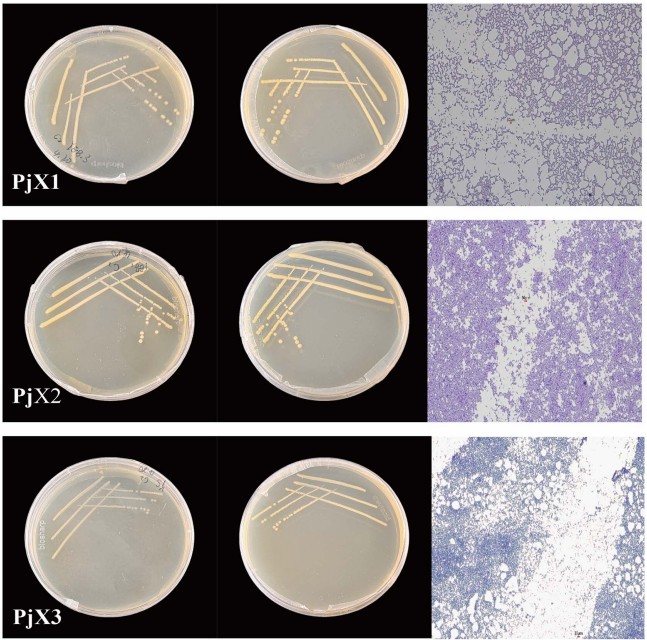

**Fig 2. Characterization of colonies and Gram staining results of endophytic bacteria *Pj*X1, *Pj*X2, and *Pj*X3 from seeds of *P. japonicus*.**

the colonies were creamy white, and the results of Gram staining were negative (Table 4), the morphology of the fungi described in Table 5.

## Molecular biological identification and homology analysis of endophyte

ITS sequence analysis indicated that the sequence lengths of seven endophytic fungi were *Pj*Z1 511 bp, *Pj*Z2 459 bp, *Pj*Z3 502 bp, *Pj*Z4 489 bp, *Pj*Z5 607 bp, *Pj*Z6 519 bp, and *Pj*Z7 512 bp, and the size of the target bands amplified by PCR using the universal primers ITS1 and ITS4 (Fig 3) agreed with the sequencing results. The 16SrDNA sequence analysis indicated that the sequence lengths of the three endophytic bacteria were *Pj*X1 1413 bp, *Pj*X2 1400 bp, and *Pj*X3 1416 bp, respectively, and the size of the target bands amplified by PCR using the universal primers 27F and 1492R (Fig 4) was consistent with the sequencing results.BLAST search was performed at NCBI and a Utilization of the software MEGA11 phylogenetic tree was constructed (Fig 5), which revealed that the sequence similarity between endophytic fungi *Pj*Z1 and accession number JX827458.1 was 100% and identified as *Fusarium tricinctum*; *Pj*Z2 and accession number MT601889.1 had 100% sequence similarity and identified as *Fusarium reticulatum*; *Pj*Z3 showed 100% sequence similarity to accession number JX401981.1 and was identified as *Fusarium tricinctum*; *Pj*Z4 showed 100% sequence similarity to accession number OR671459.1 and was identified as *Fusarium retricinctum*; *Pj*Z5 showed 96.67% sequence similarity with accession number OR543773.1 and was identified as *Fusarium tricinctum*; *Pj*Z6 showed 100% sequence similarity with

**Table 4. Morphology and Gram staining of endophytic bacteria in the seeds of *P. japonicus*.**

| Endophytic bacteria | Colony characteristics | Gram stain |
|---|---|---|
| *Pj*X1 | Rounded, with neat margins and a slightly raised center, the colonies are creamy white | Negatives |
| *Pj*X2 | Rounded, with neat margins and a slightly raised center, the colonies are creamy white | Negatives |
| *Pj*X3 | Rounded, with neat margins and a slightly raised center, the colonies are creamy white | Negatives |

**Table 5. Characteristics of endophytic fungi colonies in the seeds of *P. japonicus*.**

| Endophytic fungi | Appearance | Microscopic form |
|---|---|---|
| *Pj*Z1 | Mycelium abundant, aerial hyphae well-developed; colonies raised, tomentose, white on the front, orange on the back, with goose-yellow margins. | Mycelium septate, branched; spores green, rounded;broom-like spore-producing structures. |
| *Pj*Z2 | Mycelium fewer, aerial hyphae well developed; colonies raised, tomentose, adaxial margin yellow, middle pink, abaxial light yellow-brown, margin goose-yellow. | Mycelium septate, branched; spores green, rounded;broom-like spore-producing structures. |
| *Pj*Z3 | Mycelium abundant, aerial hyphae well-developed; colonies raised, tomentose, white adaxially, light yellow abaxially. | Mycelium septate, branched; spores green, rounded;broom-like spore-producing structures. |
| *Pj*Z4 | Mycelium abundant, aerial hyphae well-developed; colonies raised, tomentose, gray-yellow adaxially, brown abaxially, with goose-yellow margins. | Mycelium septate, branched; spores colorless, rounded;broom-like spore-producing structures. |
| *Pj*Z5 | Mycelium abundant, aerial hyphae well developed; colonies raised, cottony, white on the front, goose-yellow on the back. | Mycelium septate, branched; spores green, rounded;broom-like spore-producing structures. |
| *Pj*Z6 | Mycelium fewer, aerial hyphae well developed; colonies raised, downy, pink on the front, light orange on the back. | Mycelium septate, branched; spores green, rounded;broom-like spore-producing structures. |
| *Pj*Z7 | Mycelium abundant, aerial hyphae well-developed; colonies raised, tomentose, grayish-white adaxially, brown in the middle of the abaxial surface, grayish-brown at the edges. | Mycelium septate, branched; spores colorless, rounded;broom-like spore-producing structures. |

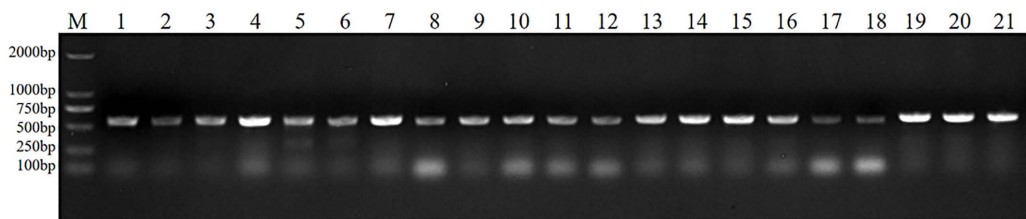

**Fig 3. PCR identification of endophytic fungi in the seeds of *P. japonicus*.** Marker:DNA Marker DL2000; 1-3 for *Pj*Z1, 4-6 for *Pj*Z2, 7-9 for *Pj*Z3, 10-12 for *Pj*Z4, 13-15 for *Pj*Z5, 16-18 for *Pj*Z6, 19-21 for *Pj*Z7.

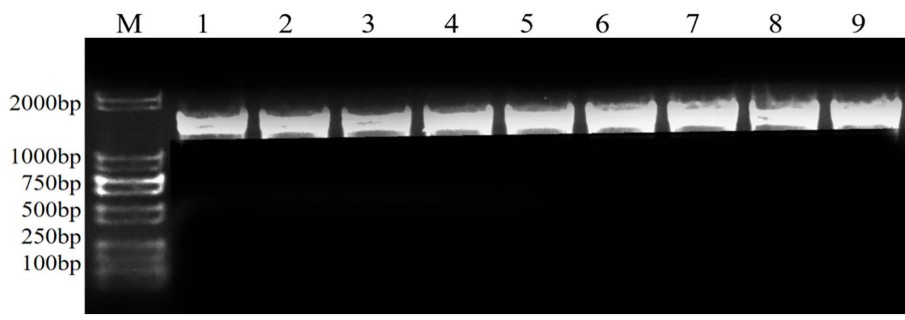

**Fig 4. PCR identification of endophytic bacteria in the seeds of *P. japonicus*.** Marker:DNA Marker DL2000; 1-3 for *Pj*X1, 4-6 for *Pj*X2, 7-9 for *Pj*X3.

accession number LT746261.1 and was identified as *Fusarium sarcochroum*; *Pj*Z7 showed 100% sequence similarity with accession number LT746261.1 with 100% sequence similarity, identified as *alternaria alternata*. endophytic bacteria *Pj*X1 with accession number JQ864378.1 with 100% sequence similarity, identified as *Enterobacteriaceae bacterium*, *Pj*X2 with accession number PQ097710.1 with 99.93% sequence similarity and identified as *Pseudomonas aeruginosa*, *Pj*X3 with 99.79% sequence similarity to accession number JX067703.1 and identified as *Enterobacteriaceae* bacterium.

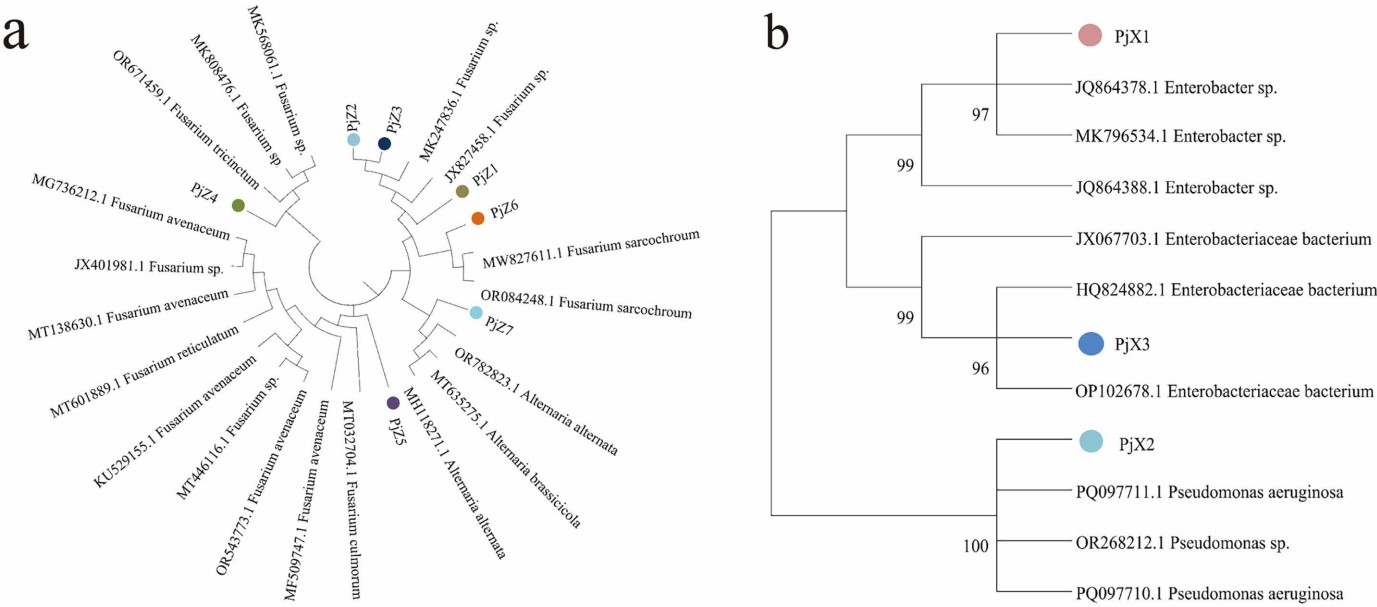

**Fig 5. Phylogenetic tree of endophytes.** a. Phylogenetic tree constructed from *PjZ1-PjZ7* rDNA-ITS sequences of the endophytic fungi from the seeds of *P. japonicus*, b. Phylogenetic tree constructed from the rDNA sequences of endophytic bacteria *PjX1-PjX3* in the seeds of *P. japonicus*.

## Determination of bacteriostatic activity of endophytic fungi

Determination of bacteriostatic activity were carried out on seven strains of endophytic fungi isolated with *Escherichia coli*, *Staphylococcus aureus* and *Bacillus subtilis*, and the results showed that (Table 6), strain *PjZ4* had inhibitory activity against *Escherichia coli*, but the inhibitory activity was weak, with an average inhibitory circle diameter of 10.48 mm, and the remaining six strains of the fungi did not have any inhibitory activity against *E. coli*; the seven strains of the endophytic fungi strains did not have any inhibitory activity against *Staphylococcus aureus*; Strain *PjZ3*, *PjZ4*, *PjZ5*, *PjZ7* had bacteriostatic activity against *B.subtilis*, strain *PjZ3* had the strongest bacteriostatic activity against *B. subtilis*, with an average inhibitory circle diameter of 23.53 mm, strain *PjZ4* had the weakest inhibitory activity against *B. subtilis*, with an average inhibitory circle diameter of 13.69 mm, and the average inhibitory circle diameters of strains *PjZ5* and *PjZ7* were respectively 16.80 mm, 17.97 mm, strains *PjZ1*, *PjZ2* and *PjZ6* had no inhibitory activity against *B. subtilis*, and no inhibitory circle was produced in the control blank drug-sensitive paper sheet (Fig 6).

**Table 6. Inhibitory activity of endophytic fungi from the seeds of *P. japonicus* against three pathogenic bacteria.**

| Strain number | *Escherichia coli (E. coli)* | *Staphylococcus aureus* | *Bacillus subtilis* |
|---|---|---|---|
| | Diameter of inhibition circle/ mm | | |
| *PjZ1* | – | – | – |
| *PjZ2* | – | – | – |
| *PjZ3* | – | – | 23.53 |
| *PjZ4* | 10.48 | – | 13.69 |
| *PjZ5* | – | – | 16.80 |
| *PjZ6* | – | – | – |
| *PjZ7* | – | – | 17.97 |

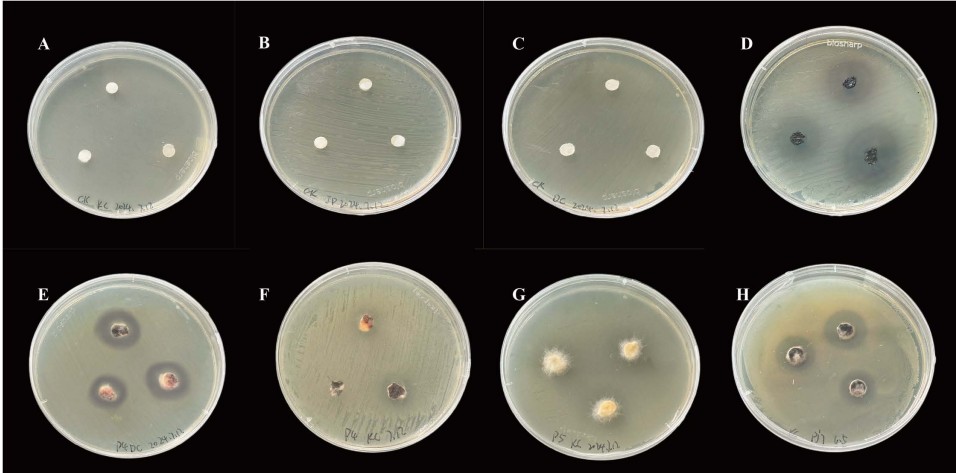

**Fig 6. Inhibitory activity of endophytic fungi from the seeds of *P. japonicus* against three pathogenic bacteria.** A, B and C are blank controls for the three pathogens; D, *Pj*3/*Bacillus subtilis*; E, *Pj*4/*E. coli*; F, *Pj*4/*B. subtilis*; G, *Pj*5/*B. subtilis*; and H, *Pj*7/*B. subtilis*.

## Bacteriostatic activity of endophyte metabolites

Antagonistic assay of the metabolites of 7 strains of endophytic fungi and 3 strains of bacteria isolated against *E. coli*, *S. aureus* and *B. subtilis* showed that (Table 7), the metabolites of the 7 strains of endophytic fungi had no inhibitory activity against *E. coli*, *S. aureus* and *B. subtilis*, and there was no inhibitory circle on the control blank drug-sensitive paper sheet as well.

Antagonistic assay of the metabolites of the three isolated endophytic bacteria with *E. coli*, *S. aureus* and *B. subtilis* revealed that the metabolites of the three endophytic bacteria had no bacteriostatic activity against *E. coli*, *S. aureus* and *B. subtilis* (Table 8).

**Table 7. Inhibitory activity of endophytic fungal metabolites from the seeds of *P. japonicus*.**

| Strain number | Escherichia coli | Staphylococcus aureus | Bacillus subtilis |
| --- | --- | --- | --- |
| *Pj*Z1 | – | – | – |
| *Pj*Z2 | – | – | – |
| *Pj*Z3 | – | – | – |
| *Pj*Z4 | – | – | – |
| *Pj*Z5 | – | – | – |
| *Pj*Z6 | – | – | – |
| *Pj*Z7 | – | – | – |

**Table 8. Inhibitory activity of endophytic bacteria metabolites from the seeds of *Panax japonicus*.**

| Strain number | *Escherichia coli* | *Staphylococcus aureus* | *Bacillus subtilis* |
| --- | --- | --- | --- |
| *Pj*X1 | – | – | – |
| *Pj*X2 | – | – | – |
| *Pj*X3 | – | – | – |

## Discussion

In recent years, the germplasm resources of *P. japonicus* have been endangered, and the conservation of the germplasm resources of *P. japonicus* has become a key issue to be solved. In the process of artificial breeding, we found that the seeds of *P. japonicus* were difficult to germinate, studies have shown that different plant seed endophytes can potentially increase the productivity of host plants [12]. However, the isolation and identification of endophytic fungi from the seeds of *Panax japonicus* have not been reported. In this study, seven strains of endophytic fungi were isolated from the seeds of *P. japonicus*, among which six strains belonged to *Fusarium,*and one strain belonged to *Alternaria alternata* The two genera of fungi had almost no inhibitory effect on *E. coli* and *S. aureus*, and had inhibitory activity on *B. cereus* but the inhibitory effect was not obvious, and the metabolites of the two genera had no inhibitory activity on the indicator organisms. The metabolites of the two genera of fungi had no bacteriostatic activity on the indicator bacteria, preliminary speculation that two genera of fungi are not plant growth promoting fungi *p.japonicus* seed germination. And many studies pointed out that *Fusarium*. and *Alternaria alternata* are pathogenic fungi that cause plant diseases, such as Yang isolation and identification of pathogenic fungi on naturally-occurring *Passiflora edulis*, and found that the pathogenic fungi that cause post-harvest *Passiflora edulis* diseases belong to *Fusarium* spp. and *Streptomyces* [13]. Zhong determined the pathogenicity of different strains of *Fusarium* species on *Nicotiana tabacum*, and showed that the pathogenic fungi of *Fusarium* root rot of *Nicotiana tabacum* were mainly shared by *Fusarium.*, and 40% of imipramine EW had an inhibitory effect on this pathogen [14]. Chen isolated and identified the pathogenic bacteria from the leaves of the diseased plants of *Paeonia lactiflorapall* ring rot, and the results showed that the pathogen of *Paeonia lactiflorapall* ring rot was *Alternaria alternata* [15]. All these studies confirmed that *Fusarium*. and *Alternaria alternata*. are not growth-promoting bacteria in plants still, both of them cause different diseases and inhibit the normal growth and development of plants in various degrees. Therefore, it is likely that the difficulty in seed germination of *P. japonicus* is due to the action of these two endophytic fungi, and further research is needed to find antibiotics to inhibit the growth of these endophytic fungi to solve the problem of the difficulty in seed germination of *P. japonicus*.

In this study, three strains of endophytic bacteria were also isolated from the seeds of *P. japonicus*, two of which belonged to *Enterobacteriaceae* and one belonged to *Pseudomonas aeruginosa*, and the metabolites of the two genera of bacteria had no inhibitory activity against the indicator bacteria, which may be due to the high concentration of the indicator bacteria or the low activity of the metabolites. Some studies have shown that *Enterobacteriaceae* has a better effect in promoting plant growth, such as Zhao isolated endophytic *Enterobacteriaceae* from the *Paspalum vaginatum* as a material to explore the germination of *Cynodon daclylon* seeds and salt tolerance after growing into a lawn, the results of the seed germination rate and salt tolerance after growing into a lawn has been improved [16]. Some scholars isolated endophytic bacteria from the root system of cucumber and found that *Enterobacteriaceae* and *Pseudomonas spp*. played an important role in the control of cucumber wilt disease [17]. Ma initially identified eight strains of bacteria isolated from plant roots and investigated the growth-promoting effects of endophytic bacteria on *ryegrass*, which showed that all eight strains of bacteria belonged to *Pseudomonas aeruginosa*. and could effectively promote the growth and development of *Lolium perenne* [18]. The results showed that all eight strains of bacteria belonged to *Pseudomonas* spp. From the results of many studies, it can be inferred that *Enterobacteriaceae* and *Pseudomonas aeruginosa*. are the growth-promoting endophyte in promoting plant growth and development, therefore, further investigation can be carried out in the future seed germination and disease control of *P. japonicus.*And it is expected to provide some ideas for the production and application of *P. japonicus.*

## Conclusion

The orthogonal experiment screened out the optimal disinfection conditions of *P. japonicus* seeds as 75% alcohol disinfection for 30s and 1% $HgCl_2$ disinfection for 9 min, which had the lowest contamination rate of 3.33%. Seven strains of endophytic fungi *PjZ1, PjZ2, PjZ3, PjZ4, PjZ5, PjZ6, PjZ7*, and three strains of *endophytic bacteria PjX1, PjX2, PjX3* were isolated and screened from the seeds of *P. japonicus*, and the strains of *PjZ1, PjZ3, PjZ4, PjZ5* were identified as

the fungi of the *Fusarium tricinctum*, and the strains of *PjZ2* was identified as the fungi of the *Fusarium* genus. *Fusarium reticulatum*, *PjZ6* is *Fusarium sarcochroum*, *PjZ7* is *Alternaria alternata*; *PjX1* and *PjX3* are *Enterobacteriaceae bacterium*, and *PjX2* is *Pseudomonas aeruginosa*. The results of inhibition experiments showed that only *PjZ4* had inhibitory activity against *Escherichia coli*, *PjZ3*, *PjZ4*, *PjZ5*, and *PjZ7* had inhibitory activity against *B. subtilis*, and none of the 7 endophytic fungi strains had inhibitory activity against *Staphylococcus aureus*, The experiment showed that the metabolites of 10 strains of endophytic strains had no bacteriostatic activity against *Escherichia coli*, *Staphylococcus aureus* and *Bacillus subtilis*. After preliminary judgment, the isolated *Fusarium spp.* and *Alternaria alternata.* are not the growth-promoting endophytes in the growth and development process of *P. japonicus* seed, and it can be inferred that the difficulty in seed germination of *P.japonicus* may be caused by the growth of these two genera of fungi inhibiting the synthesis of growth hormone by the seeds during the seed germination process. The isolated *Enterobacteriaceae bacterium* and *Pseudomonas spp.* are the growth-promoting bacteria in plant growth and development, so these two genera of bacteria can be utilized in the cultivation process of *P. japonicus* to promote its growth and development. The endophytes isolated in this study provide a theoretical basis for the screening of endophyte antagonist strains of *P. japonicus* and for seed germination and disease control of *P. japonicus*.The antibacterial activity results are modest,for fungus showed weak inhibition, and none of the metabolites did. No functional testing was done to prove that these endophytes help or harm seed germination directly. It is more of a foundational study, setting the stage for future work.Next, we will further apply the isolated endophytic bacteria to seeds and test their effect on seed germination.

## Supporting information

**S1 File. Raw images.**
(PDF)

**S2 File. Dataset.**
(XLSX)

## Acknowledgments

We would like to thank Professor Lai Zhang for his careful guidance of this study, School of Life Science, Guizhou Normal University and the Guizhou Provincial Innovation Center for Mountainous and Efficient Agriculture for their strong support. We thank the editors and reviewers for their hard work on our papers.

## Author contributions

**Conceptualization:** Rui Jin, Lai Zhang.

**Data curation:** Rui Jin.

**Funding acquisition:** E. Liang, Lai Zhang.

**Methodology:** Rui Jin, Xilun Huang, Juan Huang.

**Resources:** Lai Zhang.

**Software:** Juan Huang.

**Supervision:** E. Liang, Lai Zhang.

**Validation:** Tingting Tang, Juan Huang.

**Visualization:** Tingting Tang, Xilun Huang.

**Writing – original draft:** Rui Jin, Tingting Tang.

**Writing – review & editing:** Lai Zhang.

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
