## [Decision Letter · Decision Letter 0]

7 Jan 2025

Isolation, identification and antibacterial activity of endophytic bacteria from the seeds of < Panax japonicus>

PLOS ONE

Dear Dr. Zhang,

Thank you for submitting your manuscript to PLOS ONE. After careful consideration, we feel that it has merit but does not fully meet PLOS ONE’s publication criteria as it currently stands. Therefore, we invite you to submit a revised version of the manuscript that addresses the points raised during the review process.

We look forward to receiving your revised manuscript.

Kind regards,

Abhay K. Pandey

Academic Editor

PLOS ONE

“Science and Technology Funded Project of Guizhou Provincial Science and Technology Department (Guizhou Science and Technology Foundation -ZK Grant NO.(2023) Key 002).

National Natural Science Foundation of China (Grant NO.31660252).

Guizhou Provincial Department of Education Youth Science and Technology Talent Development Project (Qian Jiao He KY character Grant NO.[2022]029).

Anshun College graduate Research Innovation project asxyyjscx Grant NO.202410.”

“We would like to thank Professor Lai Zhang for his careful guidance of this study and the Guizhou Provincial Innovation Center for Mountainous and Efficient Agriculture for its strong support.

This work was supported by the four project grants:

Science and Technology Funded Project of Guizhou Provincial Science and Technology Department (Guizhou Science and Technology Foundation -ZK Grant NO.(2023) Key 002).

National Natural Science Foundation of China (Grant NO.31660252).

Guizhou Provincial Department of Education Youth Science and Technology Talent Development Project (Qian Jiao He KY character Grant NO.[2022]029).

Anshun College graduate Research Innovation project asxyyjscx Grant NO.202410.

We thank the editors and reviewers for their hard work on our papers.”

“Science and Technology Funded Project of Guizhou Provincial Science and Technology Department (Guizhou Science and Technology Foundation -ZK Grant NO.(2023) Key 002).

National Natural Science Foundation of China (Grant NO.31660252).

Guizhou Provincial Department of Education Youth Science and Technology Talent Development Project (Qian Jiao He KY character Grant NO.[2022]029).

Anshun College graduate Research Innovation project asxyyjscx Grant NO.202410.”

6. Please remove your figures from within your manuscript file, leaving only the individual TIFF/EPS image files, uploaded separately. These will be automatically included in the reviewers’ PDF.

7. Please include your tables as part of your main manuscript and remove the individual files. Please note that supplementary tables (should remain/ be uploaded) as separate "supporting information" files.

Reviewers' comments:

Reviewer's Responses to Questions

**Comments to the Author**

1. Is the manuscript technically sound, and do the data support the conclusions?

Reviewer #1: Yes

Reviewer #2: Partly

2. Has the statistical analysis been performed appropriately and rigorously?

Reviewer #1: No

Reviewer #2: N/A

3. Have the authors made all data underlying the findings in their manuscript fully available?

Reviewer #1: Yes

Reviewer #2: Yes

4. Is the manuscript presented in an intelligible fashion and written in standard English?

Reviewer #1: Yes

Reviewer #2: No

Reviewer #1: This manuscript presents a study on the isolation, identification, and antibacterial activity of endophytic fungi and bacteria from Panax japonicus seeds. The topic is relevant and potentially significant for understanding seed biology and developing biocontrol agents. The study is generally well-structured, but there are several areas that require clarification and improvement before publication. Overall, the results are preliminary and lack strong evidence, but still valuable for further research.

Abstract (Lines 39-65):

• Line 40: Consider stating the specific aim of this study more clearly, such as: "The aim of this study was to isolate..."

• Line 48: It is confusing to read "Seven endophytic fungi strains were isolated" immediately followed by mentioning that "PjZ1, PjZ3, PjZ4 and PjZ5 were identified as Fusarium". This makes it appear that only these four were fungi and not all 7 strains. You should rephrase this to clarify that ALL 7 are fungal.

• Line 55: Instead of "Only PjZ4 had antibacterial activity", it may be more appropriate to say, "Among the isolates, only PjZ4 showed significant antibacterial activity...". This highlights that the other fungal isolates did have some inhibition on Bacillus subtilis, even if not significant.

• Line 58: Conclusion: It is advised to have a stronger conclusion. For instance, the authors could summarize the main findings, including the genus level of the identified strains, and highlight the significance of their study for future research.

• Line 63: The keywords could be improved by making them more specific. For example, replace "Antibacterial activity" with "Endophytic antibacterial activity".

Introduction (Lines 66-81):

• Line 68: The phrase "also known as Disporopsis fuscopicta Hance" is an unnecessary detail here as it is not the same genus.

• Line 70: Should add citations to support that "Pharmacological studies have shown that Panax japonicus..."

• Line 75: The statement regarding lack of reports on endophytic fungi/bacteria from seeds is good, but there needs to be something to indicate why this is valuable. For example, why are seeds and what importance is this to P. japonicus?

• Line 77-80: The end of the introduction, there should be something to emphasize the biological functions of the bacteria isolated.

Materials and Methods (Lines 82-162):

• Line 86: Provide the geographic coordinates or location details of the experimental site.

• Line 90: The composition of NA (Nutrient Agar) should be described better. The recipe is given as "Peptone 10g, Beef Paste 3g, Sodium Chloride 5g, Agar 20g, distilled water 1000mL, pH:7.2." It would be better practice to say "Beef Extract" instead of "Beef Paste", as well as identify if peptone is tryptone or another form.

• Line 103: Provide a citation or additional details to justify the use of this orthogonal testing method.

• Line 107-109: It is difficult to follow that section. Re-word and clarify that after disinfection, the seeds were plated to analyze contamination rate before proceeding with isolation.

• Line 115: Describe the sterile filter paper with more details, such as pore size and manufacturer.

• Line 117: Clarify why the seed is cut in half before plating.

• Line 119: What is meant by "seed preservation" here? It is better to say for "preservation of the isolated strains"

• Line 127-136: The morphological observation section lacks clarity. How many samples were observed per isolate? Were observations done in replicates? More detail on how results were "recorded" should be included.

• Line 137-139: The DNA extraction procedure needs to be described with more detail: What is the kit that was used? Also, it is better to use "according to the manufacture's instructions" rather than just stating "according to the instructions".

• Line 144: The reaction volume and composition is confusing. It would be better to say "a 50uL reaction volume containing..." instead.

• Line 151: The statement about the primer name is inconsistent, the previous section mentioned primer ITS1 and ITS4 but now are described as "primers 27F (5'-AGAGTTTGATCCTGGCTCAG-3') and 1492R (5'-TACGGCTACCTTGTTACGAC".

• Line 157: What percentage of agarose gel was used in gel electrophoresis?

• Line 166: Please add more detail about the selection of pure isolates. For instance, whether single colonies were used to prepare the pure isolates.

• Line 167: What does “adjusted to OD600=0.2 in 0.85% saline” means? Also, how was the 0.2 OD reading achieved?

• Line 174: It is good to mention the preparation of fungal cake, but describe the type of agar that was used.

• Line 178: Please mention which drug-sensitive paper was used.

Results (Lines 163-205):

• Line 215: The abbreviation ANOVA was already introduced, so no need to repeat it again.

• Line 222-229: It would be better to include a table of the Gram staining morphology results for better data presentation.

• Line 242: Avoid the use of the abbreviation for "16S ribosomal DNA", write as "16S rDNA" for consistency.

• Line 242-261: The method of identifying the endophytes is unclear. You should state the database that was used, such as NCBI (National Center for Biotechnology Information). Also, it would help to give more details regarding the process of phylogenetic tree construction.

• Line 276: Clarify that “Antagonistic experiments” are referring to the testing of antibacterial activity. Also, clarify that the bacteria used in the antagonistic experiments were pathogenic.

• Line 281-284: Describe the statistical significance of the differences observed.

• Line 298-307: The lack of activity should be presented in a table.

Discussion (Lines 308-376):

• Line 310: It is good that the authors note the potential of endophytes in productivity, but needs to be better linked to this study.

• Line 313: Clarify why the presence of Fusarium and Alternaria is important to discuss.

• Line 315: Is this significant? What do the non-inhibitory results suggest?

• Line 320: Again, why is it important to note the dominant fungi were not found?

• Line 330: Pseudocercospora variicola is a fungal species, but is treated as bacterial here. This section would benefit from being clarified, and better linked to the results, that this particular study may be considered a contaminant.

• Line 335: The authors should further discuss the importance of their finding in P. japonicus.

• Line 337-344: Needs a better transition to the importance of Pseudomonas and Enterobacteriaceae.

• Line 361-365: Reiterate more clearly. The authors should briefly list the identified species, but also highlight the dominant species in the study.

• Line 366-376: The conclusion is not comprehensive enough and should be revised to clearly state the main outcomes, limitations, and implications for future studies.

• Statistical Analysis: The manuscript briefly mentions statistical analysis (ANOVA) but more detailed information about the statistical methods used should be included. The data presentation could be enhanced by adding mean values, standard deviation values, and P values as appropriate.

• Language: The manuscript needs careful proofreading. There are several instances of grammatical errors and awkward phrasing that need to be revised.

Reviewer #2: The authors attempted to isolate and identify bacterial and fungal endophytes and studied the antibacterial activity of endophytic isolates from Panax japonicus. As a whole, the research findings may be helpful to other researchers. However, the scientific statements made in the MS are not clear, and the information flow is not up to the mark.

For example, the paper's title is “Isolation, identification and antibacterial activity of endophytic bacteria from the seeds of Panax japonicus." However, the MS reports both bacterial and fungal endophytes. For instance, under the abstract, the authors write, “Seven endophytic fungi strains were isolated from the seeds of Panax japonicus, named…………”

The conclusion is very weak and highlights only the isolation/identification of endophytic fungi. Innovation elements are missing from the MS. The conclusion before the references list is very weak from a scientific perspective, and many statements made don't fall under the conclusion.

The knowledge gap or the background provided in the Introduction is scientifically weak. Furthermore, the scientific arguments that are made under discussion are weak.

The authors have the results and data; however, the overall quality of the MS needs to be improved.

**Do you want your identity to be public for this peer review?** For information about this choice, including consent withdrawal, please see our Privacy Policy

Reviewer #1: No

Reviewer #2: No

---

## [Author Response · Author response to Decision Letter 1]

16 Jan 2025

Response to Reviewers

Dear editors and reviewers,

Thank you for your comments on the title of our manuscript 'Isolation, identification and antibacterial activity of endophyte from the seeds of Panax japonicus '. Your comments will help us to revise and improve the manuscript, and will also play an important guiding role in our future research in the Panax japonicus or other fields. We are very grateful to you for your comments and comments on the content of our manuscripts, especially for the questions we have pointed out in words. We have carefully studied these comments and tried to modify the manuscript, hoping to get your approval. Revisions are marked in red in the manuscript. The following is a response to your comments.

Editor

1.Please ensure that your manuscript meets PLOS ONE's style requirements, including thosefor file naming.

Response: Dear Editor, thank you for your reminder that we have modified the format and naming according to the journal requirements.

2.Please state what role the funders took in the study.

Response:Thanks to the editors for being so careful, we have clarified in the Cover letter: “The above four project funds provide financial support and project guidance for this research.”

3.We note that you have provided funding information that is currently declared in your Funding Statement. However, funding information should not appear in the Acknowledgments section or other areas of your manuscript. We will only publish funding information present in the Funding Statement section of the online submission form.Please remove any funding-related text from the manuscript and let us know how you would like to update your Funding Statement.

Response:Thanks for the editor's reminder and suggestion, we have made modifications, deleted the text related to funding from the manuscript, and included the funding information in the Cover letter.

4.When completing the data availability statement of the submission form, you indicated that you will make your data available on acceptance. We strongly recommend all authors decide on a data sharing plan before acceptance, as the process can be lengthy and hold up publication timelines. Please note that, though access restrictions are acceptable now, your entire data will need to be made freely accessible if your manuscript is accepted for publication. This policy applies to all data except where public deposition would breach compliance with the protocol approved by your research ethics board. If you are unable to adhere to our open data policy,please kindly revise your statement to explain your reasoning and we will seek the editor's input on an exemption. Please be assured that, once you have provided your new statement, the assessment of your exemption will not hold up the peer review process.

Response:Thanks for the editor's suggestion, we have started to prepare the work of data uploading and data sharing, and we promise to do our best to complete the work before the manuscript is accepted.

5.PLOS ONE now requires that authors provide the original uncropped and unadjusted images underlying all blot or gel results reported in a submission’s figures or Supporting Information Files.In your cover letter, please note whether your blot/gel image data are in Supporting Informationor posted at a public data repository, provide the repository URL if relevant, and provide specificdetails as to which raw blot/gel images, if any, are not available. Email us at plosone@plos.org if you have any questions.

Response:Thanks for the editors' valuable feedback and suggestions on the chart information. We have put the unadjusted chart into the file separately and explained it in the Cover letter: we will agree that blot/gel image data are in Supporting Information or posted at a public data repository.

6.Please remove your figures from within your manuscript file, leaving only the individual TIFF/EPS image files, uploaded separately ,and please include your tables as part of your main manuscript and remove the individual files.

Response:Thanks to the careful reminding of the editor, we have uploaded the single TIFF/EPS image file and deleted the chart from the manuscript, and the single table file has also been deleted.

Reviewer 1

Abstract (Lines 39-65):

• Line 40: Consider stating the specific aim of this study more clearly, such as: “The aim of this study was to isolate...”

Response:We agree with the reviewer on this point, and we have revised it in the manuscript: “ The purpose of this study is to isolate and identify endophytic bacteria from the seeds of Panax japonicus, finding a dominant strain that favors seed germination in Panax japonicus and to provide theoretical reference for further research on the effect of endophytic bacteria on seed germination. ”

• Line 48: It is confusing to read "Seven endophytic fungi strains were isolated" immediately followed by mentioning that "PjZ1, PjZ3, PjZ4 and PjZ5 were identified as Fusarium". This makes it appear that only these four were fungi and not all 7 strains. You should rephrase this to clarify

that ALL 7 are fungal.

Response:We thank the reviewer for making this important point, which we have elaborated in the manuscript: “Seven endophytic fungi strains were isolated from the seeds of Panax japonicus, named PjZ1, PjZ2, PjZ3, PjZ4, PjZ5, PjZ6, and PjZ7, respectively. Four strains belonged to the genus Fusarium tricinctum, one to Fusarium reticulatum, one to Fusarium sarcochroum, and one to Alternaria alternata”

• Line 55: Instead of "Only PjZ4 had antibacterial activity", it may be more appropriate to say,"Among the isolates, only PjZ4 showed significant antibacterial activity...". This highlights that the other fungal isolates did have some inhibition on Bacillus subtilis, even if not significant.

Response:We thank the reviewer for pointing out a more appropriate statement for us, and we have readjusted it: “Among the endophytic fungal strains isolated, only PjZ4 showed significant bacteriostatic activity against Escherichia coli. ”

• Line 58: Conclusion: It is advised to have a stronger conclusion. For instance, the authors could summarize the main findings, including the genus level of the identified strains, and highlight the significance of their study for future research.

Response:Thanks to the reviewer for his valuable suggestions. In view of the suggestions, we have made a new explanation after careful consideration:“In this study, we investigated the method of bamboo ginseng seed disinfection, and the results showed that 75% alcohol disinfection for 30s and 1% HgCl2 disinfection for 9min resulted in the lowest contamination rate of 3.33%, in which the greatest influence on the seed disinfection effect was the concentration of HgCl2. Meanwhile, 11 endophyte species of Panax japonicus seeds were isolated and identified, Fusarium spp. and Alternaria alternata. were not the dominant species in the growth and development process of Panax japonicus while Enterobacteriaceae and Pseudomonas sp. were the dominant species in the promotion of plant growth and development, and provides a theoretical reference for further research on the biological functions and active substances of endophytic fungi in the seeds of Panax japonicus.”

• Line 63: The keywords could be improved by making them more specific. For example, replace "Antibacterial activity" with "Endophytic antibacterial activity".

Response:Thanks for your helpful suggestions, we have updated the keyword "Antibacterial activity" to "Endophytic antibacterial activity"

Introduction (Lines 66-81):

• Line 68: The phrase "also known as Disporopsis fuscopicta Hance" is an unnecessary detail here as it is not the same genus.

Response:We appreciate your reminder that we have removed the word from the manuscript.

• Line 70: Should add citations to support that "Pharmacological studies have shown that Panax japonicus..."

Response:We are grateful for your reminder that in the manuscript, our citation for this part reads [2-4]:[2]SHEN Peiyao, ZHANG Ye, LI Yuqin, YAN Jianye, WANG Yuanqing. Research progress on biological activity and quality control of Panacis Japonici Rhizoma[J]. Food and Machinery,2021,37(11):211-220. [3]GUO Zhe, FENG Zhitao, ZHANG Haoran, YAN Lan, LIANG Mingge, MEI Zhigang, CAI Sanjin. Progress of research on Panax japonicus and its preparation for the treatment of rheumatoid arthritis[J]. Chinese Materia Medica,2019,42(04):941-944. [4]LI Chunyan, ZHANG Jie, LI Jinping, et al. Research on Chemical Constituents and Biological Activities of Rhizoma of Panax japonicus[J]. Introduction to Traditional Chinese Medicine,2012,18(04):68-71.DOI:10.13862/j.cnki.cn43-1446/r.2012.04.050.

• Line 75: The statement regarding lack of reports on endophytic fungi/bacteria from seeds is good, but there needs to be something to indicate why this is valuable. For example, why are seeds and what importance is this to P. japonicus? And Line 77-80: The end of the introduction, there should be something to emphasize the biological functions of the bacteria isolated.

Response:We would like to thank the reviewer for pointing out this problem. The clarification of this problem will make the purpose of this paper more clear. Therefore, we have further elaborated this part in the manuscript:“Because of the high efficacy and high price of the rare wild Chinese herbal medicine, it is almost difficult to find wild resources of Panax japonicus in the wild, so the artificial propagation of Panax japonicus seeds is urgent.Therefore, the selection of Panax japonicus seeds, leaves and stems for seedling propagation is a better choice of explants. It is noteworthy that the seeds of Panax japonicus have the phenomenon of after-ripening, and the contamination rate is high during the germination process of histoculture, which makes the germination difficult. Therefore, we carried out further exploration. It has been shown that endophytes of medicinal plants are important for promoting the growth of medicinal plants, improving the quality of medicinal plants, and eliminating the barriers of continuous cultivation.”

Materials and Methods (Lines 82-162):

• Line 86: Provide the geographic coordinates or location details of the experimental site.

Response:We would like to thank the reviewer for suggesting that our detailed address is"Guizhou Innovation Center of Mountain Specialty and Efficient Agriculture" of Anshun College, Anshun City, Guizhou Province, China, has been presented in the manuscript.

• Line 90: The composition of NA (Nutrient Agar) should be described better. The recipe is given as "Peptone 10g, Beef Paste 3g, Sodium Chloride 5g, Agar 20g, distilled water 1000mL, pH:7.2." It would be better practice to say "Beef Extract" instead of "Beef Paste", as well as identify if peptone is tryptone or another form.

Response:We would like to thank the reviewers for their careful comments on this and we have corrected "Beef Extract" to" Beef paste".

• Line 103: Provide a citation or additional details to justify the use of this orthogonal testing method.

Response:Thank you very much for your professional advice and we have added relevant citations in this part of the manuscript:[8]ZHU Lili,ZHANG Yemeng,LI Wancai,et al. Comprehensive evaluation of the effects of different disinfectants on surface disinfection and germination of quinoa seeds[J]. Journal of Biology,2023,40(4):83-89. DOI:10.3969/j.issn.2095-1736.2023.04.083.

• Line 107-109: It is difficult to follow that section. Re-word and clarify that after disinfection, the seeds were plated to analyze contamination rate before proceeding with isolation.

Response:We thank the reviewers for their important comments on this, which we have rephrased in the manuscript:“Prior to the isolation of endophytes from the seeds of Panax japonicus, the seeds of Panax japonicus were sterilized under different conditions and the contamination rate under different sterilization conditions were analyzed.”

• Line 117: Clarify why the seed is cut in half before plating.

Response:Thank you very much for your professional advice and we have added relevant citations in this part of the manuscript:[9]ZHANG Yubai, WEN Xiangshui, WANG Yumeng, et al. Isolation and characterization of endophytic fungi from the seeds of Fructus schisandrae chinensis and their fungicidal properties[J]. Chinese Materia Medica,2020,43(5):1087-1091. DOI:10.13863/j.issn1001-4454.2020.05.008.

• Line 119: What is meant by "seed preservation" here? It is better to say for "preservation of the isolated strains"

Response:We thank the reviewers for their very meticulous comments, which we have reorganized: “After the growth of mycelium, pick the mycelium for repeated isolation and purification for 2-3 times for preservation of the isolated strains.”

• Line 127-136: The morphological observation section lacks clarity. How many samples were observed per isolate? Were observations done in replicates? More detail on how results were "recorded" should be included.

Response:Thanks to the reviewer's constructive comments on this, we have re-elaborated in the manuscript: “Pick the purified fungal hyphae and make them into clinical slices, observe and record (hyphae color, hyphae morphology, spore morphology, etc.) through an optical microscope, and refer to "Fungal Identification Manual".etc. for preliminary identification of fungi. (Three strains on each medium for each strain and three strains on each medium for observation and identification).”

• Line 137-139: The DNA extraction procedure needs to be described with more detail: What is the kit that was used? Also, it is better to use "according to the manufacture's instructions" rather than just stating "according to the instructions".

Response: Thanks to the reviewer for alerting us to this problem, which we have reflected in the manuscript:“Endophytic fungal DNA was extracted according to the instructions of the fungal DNA extraction kit (Solarbio), and endophytic bacterial DNA was extracted according to the instructions of the bacterial DNA extraction kit. Subsequently, PCR amplification was performed using the DNA of each endophytic bacterium as a template, respectively.”

• Line 144: The reaction volume and composition is confusing. It would be better to say "a 50uL reaction volume containing..." instead.

Response: Thanks to the reviewer's attention to the following details, we have corrected the statement:“Endophytic fungi were amplified by PCR using the universal primers of fungi, ITS1 (5'-TCCGTAGGTGAACCTGCGG-3') and ITS4 ( 5'-TCCTCCGCTTATTGATATGC-3') as primers, and the total volume of the PCR reaction system was 50μL, containing:4μL of DNA template, 2μL of each of ITS1 and ITS4, and 2×MasterMix 25μL, and add ddH2O 17μL.”

• Line 151: The statement about the primer name is inconsistent, the previous section mentioned primer ITS1 and ITS4 but now are described as "primers 27F (5'-AGAGTTTGATCCTGGCTCAG-3') and 1492R (5'-TACGGCTACCTTGTTACGAC".

Response: Thanks to the reviewer for raising this question. The inconsistent names of primers are due to the fact that the endophytic bacteria isolated in our paper include fungi and bacteria, and the common primers of the two bacteria are different, so the names and sequences of the two primers appear in the paper.

• Line 157: What percentage of agarose gel was used in gel electrophoresis?

Response:Thanks for the reviewer's attention to this issue: In the manuscript we mentioned that the percentage of agarose gel used was 1%.

• Line 166: Please add more detail about the selection of pure isolates. For instance, whether single colonies were used to prepare the pure isolates,and Line 167: What does “adjusted to OD600=0.2 in 0.85% saline”means? Also, how was the 0.2 OD reading achieved?

Response:Thanks to the reviewers, we have described the process more clearly in the manuscript and updated the text:“The pure indicator strains (Staphylococcus aureus, Bacillus subtilis, Escherichia coli) were picked and inoculated onto LB medium by plate streaking method for overnight cultivation; Pick single colonies on LB medium again for plate streaking overnight culture; pick single colonies to 0.85% saline to prepare the bacterial solution, adjust the concentration of the bacterial solution and measured by UV-visible spectro

---

## [Decision Letter · Decision Letter 1]

14 May 2025

Dear Dr. Zhang,

Thank you for submitting your manuscript to PLOS ONE. After careful consideration, we feel that it has merit but does not fully meet PLOS ONE’s publication criteria as it currently stands. Therefore, we invite you to submit a revised version of the manuscript that addresses the points raised during the review process.

We look forward to receiving your revised manuscript.

Kind regards,

Raed Abduljabbar Haleem, Ph.D

Academic Editor

PLOS ONE

Reviewers' comments:

Reviewer's Responses to Questions

**Comments to the Author**

Reviewer #3: (No Response)

Reviewer #4: All comments have been addressed

2. Is the manuscript technically sound, and do the data support the conclusions?

Reviewer #3: No

Reviewer #4: Yes

3. Has the statistical analysis been performed appropriately and rigorously?

Reviewer #3: No

Reviewer #4: Yes

4. Have the authors made all data underlying the findings in their manuscript fully available?

Reviewer #3: No

Reviewer #4: Yes

5. Is the manuscript presented in an intelligible fashion and written in standard English?

Reviewer #3: No

Reviewer #4: Yes

Reviewer #3: The title of your study sounds interesting. However, it appeared that this work is not as well planed scientifically as well as it should have been and the manuscript was so poorly written leading to very confusing story throughout the entire manuscript.

Let me give some examples as followed.

The title doesn't specify what group of microbial endophytes that the researchers are focusing on, but the introductory part of the abstract only mentions "endophytic bacteria" as the purpose of this study (L.26-29). However, the result part of the abstract suddenly focuses on "Seven endophytic fung[al] strains" L.34 and "Three strains of endophytic bacteria" L.38 at which sum up to be "11 endophyte species" L.50 in the conclusion part of the abstract. This bring up the question of where the number 11 come from. This is on top of the inconsistent issue with the correct way to write all scientific names both in the abstract and in the rest of the manuscript.

Introduction:

L.59 I[n]troduction

The beginning of the introduction is a bit confusing and should start with "The chemical composition of dried rhizome of Panax japonicus C.A. Mey. mainly includes ..." For the rest of the introduction, any scientific name that has already been written in full previously, you should abbreviate the genus name for the subsequent use in the manuscript.

Materials and methods:

Culture medium section is very confusing. Components of some media were described in details while some appears to be side-of-the-bottle instruction on how to prepare them while many only have the names of the media mentioned without any details. Also, one medium that was mentioned in the "Disinfection" sub-section was not included L.114.

The sub-section on "Screening of seed disinfection conditions" is even more confusing because you described the "disinfection time of 6min, 12min, 18min" L.105 which make me wonder where the "9min" in the rest of your manuscript comes from.

L.126 "the same as 1.6 Seed Disinfection Method" What does 1.6 refer to?

L.129 is very confusing. What is the "and line the purification, purify 2-3 times to keep the seeds."

L.135-136 (Three strains on each medium for each strain and three strains on each medium for observation and identification). ???

Molecular biology identification:

L.142 mentioned that "Endophytic fungal DNA was extracted" but then on the same paragraph L.146 why "DNA of each endophytic bacterium as a template"? Also, the PCR conditions were described inconsistently regarding the volume of each components and there was no mention of concentration of the primers used and no reference to them.

L.178-185 were very confusing.

L.187 L.201 How did you measure the size of the ring of inhibition? Area? Circumference? Diameter?

L.210-211 A bit unclear writing.

L.215 What is 1.9.2?

Table 3 doesn't seem to has been clearly explained in text.

There are no mention to Fig.1 and Fig 2 in text.

L.282 What is "Streptomyces alternaria alternata"?

Fig.5 The phylogenetic tree reconstructions were improperly done such as lacking suitable outgroup sequences and limited reference sequences.

L.301-302 What is "endophytic fungal bacteria"? Which table or figures are the results of endophytic bacteria?

For the metabolites perhaps there is no need to tables 7 and simply report as no effect. Also, is table 8 the exact of table 7?

The discussion is misleading and not supported by your results. For example, what data from your study to support that the two genera of fungi were not the dominant species of endophytic fungi in the seeds when you found these 2 the most? What is the meaning to "dominant species"?

L.382 What is "salt tolerance after ping"?

The Conclusion is also very confusing with regard to "dominant species" and here we found "10 endophytic strains of endophytic fungi" L.407 and "10 strains of endophytic bacteria" L.409. Moreover, your study did not have any experiment on the effect of isolated microbial endophytes on the growth and development process of Panax

japonicus at all and thus appears to be over claiming.

There are so many other similar mistakes in this manuscript that I felt it would consume too much of my time to point them all out. Thus, I would suggest then you find a native English speaker to help you with proof reading it before any submission.

Reviewer #4: The study offers a useful sterilization protocol and a foundational catalog of endophytes from P.japonicus seeds, relevant to medicinal plant propagation and early-stage biocontrol exploration.

However, its scientific impact is limited by modest antibacterial results and the lack of functional validation of microbial roles. The manuscript requires structural and grammatical revisions, clearer scientific terminology, and more specific explanations to ensure clarity and relevance for a broader scientific audience. I have raised few questions and recommendation below based on overall reading/analyzing your paper.

The authors have not clearly or justifiably defined the purpose, context, and rationale of their study in a way that a broader audience, even an interdisciplinary scientific audience,can easily understand. They jump into procedural detail without first explaining why the work matters in a straightforward way. Ambiguities and missing explanation raise these question in introduction section: Why are the seeds being germinated in a lab? What problem is being solved? (The manuscript mentions contamination, but doesn't clearly connect it to seed failure, dormancy, and germination goals). Who benefits from this research? Is it for agricultural production, conservation, medicinal compound extraction? This is only vaguely implied. Why are E. coli, S. aureus, and B. subtilis used in tests? Not explained in ecological or applied terms.

Authors still did not report sterility checks (e.g., plating final rinse water), standard in seed disinfection protocols to confirm efficacy and rule out false negatives.

Why choose these test pathogens? These human pathogens are common in antagonism assays, but their relevance to P. japonicus seed health or soil microbiome context is unclear. Would plant pathogens (fro example Fusarium oxysporum, Pythium spp.) be more ecologically relevant?

The use of HgCl2 poses health and environmental risks. Recent protocols favor bleach (NaOCl) or H2O2 alternatives with comparable efficacy . Justify choice of HgCl₂ over safer agents, or test alternatives.

Antibacterial activity was only measured as inhibition zones from agar plugs and disc diffusion. For rigorous comparison,why minimal inhibitory concentrations (MICs) or broth microdilution assays were not performed?.

The discussion notes potential hormone-related inhibition by Fusarium spp., but offers no data on seed viability or germination trials post-sterilization. Incorporating a germination assay would solidify claims about contamination impact.

Have you considered testing endophyte re-inoculation on seed germination or seedling growth? Such functional assays are critical to demonstrate PGP potential and vave you deposited all sequence data in GenBank with accession numbers?

Instead of reporting what was done, the authors write Methods section as if they are giving instructions, like a lab manual.These kinds of descriptions r didactic, rather than declarative and scientific.For a research article, methods should be written in the past tense, reporting what was actually done, rather than describing steps as if in a lab manual.

The manuscript STILL REQUIRES careful revision for grammatical accuracy, sentence structure, and scientific terminology. Clearer sentence construction and restructuring of ambiguous or repetitive descriptions are recommended to improve clarity and precision. PLEASE go through the manuscript and check for grammatical/structural errors.

I CONCLUDE THESE MAJOR LIMITATION IN YOUR WORK: (Should be acknowledged in relevant section): The antibacterial activity results are modest,only one fungus showed weak inhibition, and none of the metabolites did. No functional testing was done (yet) to prove that these endophytes help or harm seed germination directly. It is more of a foundational study, setting the stage for future work (e.g., applying isolated strains to seeds and testing growth impact).

**Do you want your identity to be public for this peer review?** For information about this choice, including consent withdrawal, please see our Privacy Policy

Reviewer #3: No

Reviewer #4: **Yes: ** Saleem Ahmad

---

## [Author Response · Author response to Decision Letter 2]

3 Jun 2025

Dear editors and reviewers,

Thank you for your comments on the title of our manuscript 'Isolation, identification and antibacterial activity of endophyte from the seeds of Panax japonicus '. Your comments will help us to revise and improve the manuscript, and will also play an important guiding role in our future research in the Panax japonicus or other fields. We are very grateful to you for your comments and comments on the content of our manuscripts, especially for the questions we have pointed out in words. We have carefully studied these comments and tried to modify the manuscript, hoping to get your approval. Revisions are marked in red in the manuscript. The following is a response to your comments.

Reviewer 3

1.The title doesn't specify what group of microbial endophytes that the researchers are focusing on, but the introductory part of the abstract only mentions "endophytic bacteria" as the purpose of this study (L.26-29). However, the result part of the abstract suddenly focuses on "Seven endophytic fung[al] strains" L.34 and "Three strains of endophytic bacteria" L.38 at which sum up to be "11 endophyte species" L.50 in the conclusion part of the abstract. This bring up the question of where the number 11 come from. This is on top of the inconsistent issue with the correct way to write all scientific names both in the abstract and in the rest of the manuscript.

Response:We thank the reviewer for making this important point After rechecking, we have corrected the errors in the manuscript caused by our carelessness. A total of 10 endophyte are mentioned in the article, and we have also checked and corrected all scientific names.

2.L.59 I[n]troduction

The beginning of the introduction is a bit confusing and should start with "The chemical composition of dried rhizome of Panax japonicus C.A. Mey. mainly includes ..." For the rest of the introduction, any scientific name that has already been written in full previously, you should abbreviate the genus name for the subsequent use in the manuscript.

Response:We thank the reviewer for pointing out a more appropriate statement for us�and we have readjusted it:“Panax japonicus C.A. Meyer. is a famous folk medicine of the Tujia ethnic group, and is regarded as the “king of herbs’ by the Tujia ethnic group of Enshi Hubei. ”

3.Materials and methods:

Culture medium section is very confusing. Components of some media were described in details while some appears to be side-of-the-bottle instruction on how to prepare them while many only have the names of the media mentioned without any details. Also, one medium that was mentioned in the "Disinfection" sub-section was not included L.114.

Response:Thanks for your helpful suggestions, we have updated �“Fungi isolation medium: Potato Dextrose Agar Medium (PDA solid medium) 40.1g, distilled water 1000mL, pH:5.8. Bacterial isolation medium: Beef Extract Peptone Medium (NA solid medium): Peptone 10g, Beef Extract 3g, Sodium Chloride 5g, Agar 20g, distilled water 1000mL, pH:7.2. Experimental Medium for Fungi Inhibition Nutrient broth (NB liquid medium): peptone 10g, Beef Extract 3g, sodium chloride 5g, distilled water 1000mL, pH: 7.2, Beef Extract Peptone Medium (NA solid medium). Fungi fermentation medium: PDB liquid medium, Beef Extract Peptone Medium (NA solid medium), Nutrient Broth (NB liquid medium). PDA, NA and NB medium are prepared as above.Bacterial fermentation medium: LB solid medium: LB dry powder 25g, distilled water 1000mL, Agar 15g pH:7.0; LB liquid medium: LB dry powder 25g, distilled water 1000mL, pH:7.0; MH medium: Mueller-Hinton Agar 38g, distilled water 1000mL, pH:7.3”

4.The sub-section on "Screening of seed disinfection conditions" is even more confusing because you described the "disinfection time of 6min, 12min, 18min" L.105 which make me wonder where the "9min" in the rest of your manuscript comes from.

Response:We are grateful for your reminder that in the manuscript,I sincerely apologise for the confusion caused by our negligence.We used time intervals of 3 minutes, 6 minutes, and 18 minutes, and have corrected this in the manuscript.

5.L.126 "the same as 1.6 Seed Disinfection Method" What does 1.6 refer to?L.129 is very confusing. What is the "and line the purification, purify 2-3 times to keep the seeds.L.135-136 (Three strains on each medium for each strain and three strains on each medium for observation and identification).

Response:We would like to thank the reviewer for pointing out this problem.This is an error that we left behind when revising the first draft, and we have noted it in the manuscript�“Take fresh seeds of P. japonicus and rinse them under running water, then rinse them 1-3 times with sterile water, same as above for seed disinfection, peel the seeds and cut them in half, place them on NA medium with the cut side facing down, and place them in the incubator at 28℃ for cultivation. After the growth of colonies, the strains were purified using the streak plate method, the strains were purified 2-3 times and then preserved for isolated line purification.”

6.Molecular biology identification:

L.142 mentioned that "Endophytic fungal DNA was extracted" but then on the same paragraph L.146 why "DNA of each endophytic bacterium as a template"? Also, the PCR conditions were described inconsistently regarding the volume of each components and there was no mention of concentration of the primers used and no reference to them.

Response:Thanks to the reviewer for alerting us to this problem,We have corrected the description in the manuscript�“extracted DNA was used as a template for PCR amplification respectively.”and “The primer concentrations mentioned above were all 10 μmol.”

7.L.178-185 were very confusing.L.187 L.201 How did you measure the size of the ring of inhibition? Area? Circumference? Diameter?L.210-211 A bit unclear writing.

Response: Thanks to the reviewer for raising this question.We measured the diameter of the inhibition ring,We have corrected the original statement in the manuscript:“a perforator with a diameter of 0.7cm was used to prepare fungi chunks were prepared on the endophytic fungi colonies with a 0.7 cm diameter punch, and placed on the fungi plate containing the indicator species, and 3 chunks were inoculated equidistant in each plate”.

8.L.215 What is 1.9.2?There are no mention to Fig.1 and Fig 2 in text.

Response:Thank you very much for reviewing our manuscript so carefully.We have corrected the original statement in the manuscript:“Determination of bacteriostatic activity of endophytic bacterial metabolites was the same as above for the determination of bacteriostatic properties of endophytic fungi metabolites.”and “The external morphology and micromorphology of fungi and bacterial are shown in Fig.1 and Fig.2.”

9.For the metabolites perhaps there is no need to tables 7 and simply report as no effect. Also, is table 8 the exact of table 7?

Response:Thank you very much for reminding us of this problem,We have corrected table 8 in the manuscript.

10.L.382 What is "salt tolerance after ping"?The Conclusion is also very confusing with regard to "dominant species" and here we found "10 endophytic strains of endophytic fungi" L.407 and "10 strains of endophytic bacteria" L.409. Moreover, your study did not have any experiment on the effect of isolated microbial endophytes on the growth and development process of Panax japonicus at all and thus appears to be over claiming.

Response:Thank you very much for reminding us of this proble,In the manuscript, we replaced the concept of “dominant species “with “growth-promoting endophyte”,we have corrected the original statement in the manuscript:“Some studies have shown that Enterobacteriaceae has a better effect in promoting plant growth, such as Zhao isolated endophytic Enterobacteriaceae from the Paspalum vaginatum as a material to explore the germination of Cynodon daclylon seeds and salt tolerance after growing into a lawn, the results of the seed germination rate and salt tolerance after growing into a lawn has been improved.”This study preliminarily performed the isolation of endophytes and investigation of their antibacterial activities. In the future, we will focus on applying the isolated endophytes to the seed germination of Panax japonicus, with the aim of providing convenience for the production of Panax japonicus.

Reviewer 4

1.The authors have not clearly or justifiably defined the purpose, context, and rationale of their study in a way that a broader audience, even an interdisciplinary scientific audience,can easily understand. They jump into procedural detail without first explaining why the work matters in a straightforward way. Ambiguities and missing explanation raise these question in introduction section: Why are the seeds being germinated in a lab? What problem is being solved? (The manuscript mentions contamination, but doesn't clearly connect it to seed failure, dormancy, and germination goals). Who benefits from this research? Is it for agricultural production, conservation, medicinal compound extraction? This is only vaguely implied. Why are E. coli, S. aureus, and B. subtilis used in tests? Not explained in ecological or applied terms.

Response:We thank the reviewers for their important comments on this,we have carefully revised the article based on the issues you raised.in the manuscript, we mentioned:“Because of the high efficacy and high price of the rare wild Chinese herbal medicine, it is almost difficult to find wild resources of P. japonicus in the wild, so the artificial propagation of P. japonicus seeds is urgent.”The special growth environment of P. japonicus necessitates the preliminary seed germination in laboratories to create the specific conditions required for its cultivation,and this study aims to provide insights into the relevant issues encountered in the cultivation and production of P. japonicus.Additionally, since the citations in the manuscript involved determining the antibacterial effects of these three indicator bacteria against seed endophytes, we adopted this method to conduct a preliminary exploration of the endophytes in P. japonicus.We have also been inspired by your reminder, and we will strive to do better in our future research.

2.The use of HgCl2 poses health and environmental risks. Recent protocols favor bleach (NaOCl) or H2O2 alternatives with comparable efficacy . Justify choice of HgCl₂ over safer agents, or test alternatives.

Response:Thanks for your helpful suggestions.In the early stages, we used NaOCl for disinfection; however, it damaged the seed coat and negatively impacted seed germination. Therefore, we used HgCl2 for the experiment. In the future, we will explore greener and safer reagents for use in our research based on your suggestions.

3.Antibacterial activity was only measured as inhibition zones from agar plugs and disc diffusion. For rigorous comparison,why minimal inhibitory concentrations (MICs) or broth microdilution assays were not performed?

Response:We thank the reviewer for making this important point.Before using this bacterial solution concentration, we conducted concentration testing experiments. The results showed that the best experimental results were obtained at DO600=0.2, and we ultimately decided to use this concentration for our research.

4.Have you considered testing endophyte re-inoculation on seed germination or seedling growth? Such functional assays are critical to demonstrate PGP potential and vave you deposited all sequence data in GenBank with accession numbers?

Response:We are grateful for your reminder that in the manuscript,This study only conducted preliminary endophyte isolation and antibacterial experiments, without functional verification. We will proceed with this in the next step. In addition, we are currently uploading all sequence data .

5.Have you considered testing endophyte re-inoculation on seed germination or seedling growth? Such functional assays are critical to demonstrate PGP potential and vave you deposited all sequence data in GenBank with accession numbers?Instead of reporting what was done, the authors write Methods section as if they are giving instructions, like a lab manual.These kinds of descriptions r didactic, rather than declarative and scientific.For a research article, methods should be written in the past tense, reporting what was actually done, rather than describing steps as if in a lab manual.The manuscript STILL REQUIRES careful revision for grammatical accuracy, sentence structure, and scientific terminology. Clearer sentence construction and restructuring of ambiguous or repetitive descriptions are recommended to improve clarity and precision. PLEASE go through the manuscript and check for grammatical/structural errors.

Response:We thank the reviewer for making this important point,which we have made every effort to correct the tense and grammar based on your reminder, and we hope that this will resolve the issue.

6.I CONCLUDE THESE MAJOR LIMITATION IN YOUR WORK: (Should be acknowledged in relevant section): The antibacterial activity results are modest,only one fungus showed weak inhibition, and none of the metabolites did. No functional testing was done (yet) to prove that these endophytes help or harm seed germination directly. It is more of a foundational study, setting the stage for future work (e.g., applying isolated strains to seeds and testing growth impact).

Response:Thanks to the reviewer's suggestion,we have taken your suggestion and have added the manuscript:“The antibacterial activity results are modest,for fungus showed weak inhibition, and none of the metabolites did. No functional testing was done to prove that these endophytes help or harm seed germination directly. It is more of a foundational study, setting the stage for future work.Next, we will further apply the isolated endophytic bacteria to seeds and test their effect on seed germination.”

Again, thank you for giving us the opportunity to strengthen our manuscript with your valuable comments and queries. We have worked hard to incorporate your feedback and hope that these revisions persuade you to accept our submission.

Best regards.

Yours sincerely,

Rui Jin

---

## [Editor Report · Decision Letter 2]

1 Aug 2025

Isolation, identification and antibacterial activity of endophytes from the seeds of Panax japonicus

PONE-D-24-46415R2

Dear Dr. Zhang,

We’re pleased to inform you that your manuscript has been judged scientifically suitable for publication and will be formally accepted for publication once it meets all outstanding technical requirements.

Kind regards,

Raed Abduljabbar Haleem, Ph.D

Academic Editor

PLOS ONE
---

## [Editor Report · Acceptance letter]

PONE-D-24-46415R2

PLOS ONE

Dear Dr. Zhang,

I'm pleased to inform you that your manuscript has been deemed suitable for publication in PLOS ONE. Congratulations! Your manuscript is now being handed over to our production team.

Kind regards,

on behalf of

Dr. Raed Abduljabbar Haleem

Academic Editor

PLOS ONE